Quadriceps muscle reaction time in obese children

Guzmán-Muñoz Eduardo eguzmanm@santotomas.cl 1 2
Mendez-Rebolledo Guillermo 1
Sazo-Rodriguez Sergio 1
Salazar-Méndez Joaquín 1
Valdes-Badilla Pablo 3 4
Nuñez-Espinosa Cristian 5 6
Herrera-Valenzuela Tomas 7
1 School of Kinesiology, Faculty of Health, Universidad Santo Tomás , Talca , Chile
2 School of Pedagogy in Physical Education, Faculty of Education, Universidad Autónoma de Chile , Talca , Chile
3 Department of Physical Activity Sciences, Faculty of Education Sciences, Universidad Católica del Maule , Talca , Chile
4 Sports Coach Career, School of Education, Universidad de Viña del Mar , Viña del Mar , Chile
5 Medicine School, Universidad de Magallanes , Punta Arenas , Chile
6 Teaching and Research Assistance Center, Universidad de Magallanes , Punta Arenas , Chile
7 Department of Physical Activity, Sports and Health Sciences, Faculty of Medical Sciences, Universidad de Santiago de Chile , Santiago , Chile
Li Li
Electronic publication date: 2024 Feb 29
Publication date: 2024
Volume: 12
Electronic Location ID: e17050
Received 2023 Sep 28; Accepted 2024 Feb 13
Copyright: ©2024 Guzmán-Muñoz et al.
Copyright year: 2024
Copyright holder: Guzmán-Muñoz et al.
License: This is an open access article distributed under the terms of the Creative Commons Attribution License, which permits unrestricted use, distribution, reproduction and adaptation in any medium and for any purpose provided that it is properly attributed. For attribution, the original author(s), title, publication source (PeerJ) and either DOI or URL of the article must be cited.
License URL: https://creativecommons.org/licenses/by/4.0/

Keywords: Quadriceps, Electromyography, Childhood obesity, Body fat, Body mass index

Funding: The internal funds of the Academic Vice-Rector for Research and Postgraduate Studies of the Universidad Santo Tomás (Chile) 113200024 This research was financed by the internal funds of the Academic Vice-Rector for Research and Postgraduate Studies of the Universidad Santo Tomás (Chile) (project number 113200024). The funders had no role in study design, data collection and analysis, decision to publish, or preparation of the manuscript.

==============================
This study aimed to determine the influence of obesity, according to body mass index (BMI) and fat mass percentage, on quadriceps muscle reaction times. The study utilized a cross-sectional design. The sample size consisted of 42 schoolchildren (54.5% girls) aged 11 to 12 years old. Participant measurements included weight and height, which were used to categorize individuals based on BMI. Additionally, the electrical bioimpedance technique was employed to categorize participants based on their body fat percentage. A sudden destabilization test of the lower limb was performed to assess the reaction time of the rectus femoris, vastus medialis, and vastus lateralis muscles. The results show that overweight/obese children have a longer muscle reaction time for both the rectus femoris (β = 18.13; p = 0.048) and the vastus lateralis (β = 14.51; p = 0.042). Likewise, when the children were classified by percentage of body fat the results showed that overfat/obese children have a longer muscle reaction time for both the rectus femoris (β = 18.13; p = 0.048) and the vastus lateralis (β = 14.51; p = 0.042). Our results indicate that BMI and fat mass classification negativity alter the muscle reaction time in children. Overweight/obese or overfat/obese children showed longer reaction times in the rectus femoris and vastus lateralis muscles compared to children with normal weight. Based on these findings, it is suggested that in overweight and obese children, efforts not only focus on reducing body weight but that be complemented with training and/or rehabilitation programs that focus on preserving the normal physiological function of the musculoskeletal system.

Introduction

Overweightness and obesity are defined as abnormal and excessive accumulations of fat that can be detrimental to health and manifest in excess weight and body volume (Ng et al., 2014; World Health Organization, 2020). In the childhood population, obesity has become increasingly prevalent worldwide, with over 340 million children and adolescents classified as overweight or obese in 2016 (World Health Organization, 2020). According to the information presented by the World Atlas of Obesity, the trend in childhood obesity shows that by 2025 it is expected that 10% of girls and 14% of boys will be obese (World Obesity Federation, 2022). This tendency is concerning since childhood obesity is linked to an increased risk of developing chronic diseases, including cardiovascular disease, type 2 diabetes, and specific types of cancer in later stages of life (Biro & Wien, 2010).

Regarding physical performance, obesity seems to be a factor that reduces the motor efficiency of obese subjects when performing movements in a bipedal posture, presumably due to movement restrictions caused by sensorimotor alterations (King et al., 2012; Zacks et al., 2021). It has been proposed that the accumulation of fatty tissue around and within the muscle could alter the standard mechanisms of motor responses due to physiological and neuromuscular changes in the motor unit (Pajoutan, Ghesmaty Sangachin & Cavuoto, 2017). In this context, research has observed that individuals who are overweight or obese often exhibit deficiencies in both anticipatory and compensatory muscular responses (Guzmán-Muñoz et al., 2018). This would explain the low performance observed in these persons during motor skills such as postural balance, gait, and jump (Blakemore et al., 2013; DuBose et al., 2018; Guzmán-Muñoz et al., 2018; Guzmán-Muñoz et al., 2023).

The most widely used method to assess neuromuscular control is surface electromyography (sEMG). This allows detection and analysis of the electrical signal generated when a muscle contracts (Al-Ayyad et al., 2023; Guzmán-Muñoz & Méndez-Rebolledo, 2019). Among the variables that can be addressed with sEMG analysis is the muscle reaction time (also known as absolute latency), defined as the time it takes for the muscle to activate about a specific mechanical event, such as an unpredictable destabilization (Cools et al., 2003; Méndez-Rebolledo et al., 2015). It has been seen that the delay in muscle reaction times is related to a higher injury risk (De Sire et al., 2021), musculoskeletal pathologies (Méndez-Rebolledo et al., 2015), and lower motor performance (Moscatelli et al., 2016).

The knee joint is essential for the function of the lower limb in children (Flandry & Hommel, 2011). It has been suggested that the knee joint is fundamental in the function of the lower limbs in children and may be one of the joints most affected by obesity due to excess weight (Chen et al., 2020). At the knee joint level, the quadriceps muscle group plays a crucial role in the knee and the lower limb function (Madeti, Chalamalasetti & Pragada, 2015). The quadriceps muscle extends the leg at the knee joint and flexes the thigh at the hip joint. It plays a key role in everyday activities like climbing stairs, rising from a chair, running, cycling, or jumping (Madeti, Chalamalasetti & Pragada, 2015). Appropriate and timely neuromuscular control of the quadriceps muscle is essential for preserving joint health and developing motor skills, enabling children with obesity to participate actively in physical activities and sports (Guzmán-Muñoz, Valdés-Badilla & Castillo-Retamal, 2021). The adequate muscle reaction time following a sudden destabilization of a joint has been reported to be approximately 60 to 70 ms (Aruin & Latash, 1995; Nashner, Woollacott & Tuma, 1979).

Alterations in muscle reaction time have been identified in adults with previous obesity (Amiri et al., 2015; Mendez-Rebolledo et al., 2019). For example, it has been seen that overweight and obese people have slower responses than normal-weight persons in static activities (i.e., shoulder flexion) (Mendez-Rebolledo et al., 2019) and dynamic activities (i.e., walking) (Amiri et al., 2015). Few studies have analyzed changes in motor behavior through sEMG, especially in children (Blakemore et al., 2013; Hills & Parker, 1993). Blakemore et al. (2013) showed that during walking, overweight children had different muscle activation patterns than normal-weight children, which could negatively influence functionality, acquisition of motor skills, and injury risk. Despite these results, little research addresses the neuromuscular behavior of overweight and obese people with sEMG analysis, especially in children. Therefore, our study aimed to determine the influence of obesity, according to body mass index (BMI) and fat mass percentage, on quadriceps muscle reaction times. We hypothesize that those categorized as overweight/obese and overfat/obese will have delayed muscle reaction times.

Materials & Methods

Design

The study utilized a cross-sectional design and followed the guidelines outlined in the STROBE statement (Cuschieri, 2019). The participants were evaluated in a 30-minute session in a room at 21 °C in the presence of their parents and/or guardians. Participants wore shorts and were barefoot during the tests. The evaluations were performed on BMI, fat mass percentage, and muscular reaction time.

Sample size calculation

The sample size was determined based on the mean difference in the amplitude of the electromyographic signal of the rectus femoris muscle, as observed in a comparative study involving adolescents of normal weight and those who are obese while walking (De Carvalho, Martins & Teixeira, 2012). The study suggests a substantial mean difference of 10.56% with an effect size of 0.75 (Cohen’s d) between these two groups. Utilizing this information, the sample size for the current research was computed to encompass 40 participants. This calculation incorporated a significance level of 0.1 and a statistical power of 85%, ensuring the study’s ability to detect meaningful effects.

Participants

The sample size consisted of 42 schoolchildren (54.5% girls) aged 11 to 12 years old (girls, age: 11.55 ± 0.41 years; body mass: 47.75 ± 11.57 kg; height: 1.48 ± 0.05 m; and boys, age: 11.61 ± 0.44 years; body mass: 44.62 ± 8.69 kg; height: 1.45 ± 0.05 m), who attended a public educational institution in the city of Maule, Chile. Participants were selected under a non-probabilistic convenience sampling. The exclusion criteria for participants were as follows: (a) individuals displaying neurological abnormalities, (b) those who experienced musculoskeletal injuries in the lower limb, including fractures, sprains, dislocations, or muscle tears within six months before the assessments, (c) the presence of any inflammatory or painful conditions during the assessments in the lower limb, and (d) dependence on walking aids or assistive devices. In line with the Declaration of Helsinki’s principles, the participants and their parents actively granted informed consent by signing a consent form. The study received ethical approval from the local Ethics Committee (Universidad Santo Tomás, Chile), identified by registration number 13320.

Body mass index

During the assessments, the participants were asked to wear appropriate attire for measuring their body weight and standing height (Shorts, light t-shirt, and bare feet). Measurements were taken using a digital scale (Omron HBF-375 Karada Scan, Japan; accuracy of 0.1 kg) and a stadiometer (Seca model 220, Germany; accuracy of 0.1 cm). Subsequently, the BMI was calculated by dividing the body weight in kilograms by the square of the height in meters (kg/m2). BMI categories, namely normal-weight, overweight, and obese, were determined based on the BMI values and the standard deviations provided by the World Health Organization. This process requires knowing the precise age of the children in years and months, thus allowing us to identify the standard deviation range in which their BMI aligns within these parameters. Specifically, children were classified as normal-weight if their BMI fell between −1.0 and +0.9 SD, overweight between +1 and +1.9 SD, and obese if it was equal to or greater than +2.0 SD (De Onis & Lobstein, 2010).

Fat mass percentage

Regarding fat mass percentage, the electrical bioimpedance technique was used through the Omron HBF-375 body fat analyzer (Omron HBF-375 Karada Scan; Omron, Kyoto, Japan). This technique was chosen because its validity and applicability in epidemiological studies have been demonstrated, and it is recommended within the methods for studying the percentage of fat mass in children (Trang et al., 2019). For this measurement, the instructions in the manual for this equipment were followed, which have been described in a previous study (Loenneke et al., 2013). The fat mass percentage will be classified using percentile scores for sex and age based on the findings by McCarthy et al. (2006) into normal (2nd–85th percentile), overfat (>85th–95th percentile), and obese (>95th percentile).

Muscle reaction time

The skeletal muscle’s electrical signal was acquired using a Delsys electromyograph model TrignoTM Wireless sEMG System (Delsys Inc., Boston, MA, USA). Signal acquisition was performed using Discover 1.5.0 software (Delsys Inc., Boston, MA, USA). The acquired signal underwent bandpass filtering (fourth order, zero delay, Butterworth filter with frequencies ranging from 20 to 450 Hz) and was digitally amplified with a gain of 300. The system had a standard mode rejection ratio (CMRR) of >80 dB, and the signal noise level was of <0.75uV RMS. A sampling rate of 2,000 Hz was used to store the signal in the computer, employing a 16-bit resolution analog-to-digital converter (Méndez-Rebolledo et al., 2015).

A muscle reaction time test was conducted for the rectus femoris, vastus medialis, and vastus lateralis muscles. The electrodes were placed longitudinally to the muscle fibers following the guidelines of Surface Electromyography for the Non-Invasive Assessment of Muscles (SENIAM) (Hermens et al., 2000) (Table 1). Once the electrodes were positioned, the lower limb sudden destabilization test was conducted. Participants were instructed to stand on a tilt platform consisting of two separate bases, each supporting one foot arranged parallel to the other. Before the trials, they were taught to equally distribute body weight on each limb with the use of scales. The limb being evaluated was placed on the mobile base, which was remotely tilted at a 30° angle relative to the horizontal and at a velocity of ∼450°/s. A triaxial accelerometer (Delsys Inc., Boston, MA, USA) was placed on the mobile ground to accurately detect the moment of disturbance, allowing for the determination of the muscle activation time in response to the destabilizing movement caused by the tested limb. The accelerometer signal was acquired using a Delsys electromyograph model TrignoTM Wireless sEMG System (Delsys Inc., Boston, MA, USA) and signal acquisition was performed using Discover 1.5.0 software (Delsys Inc., Boston, MA, USA). The accelerometer signal was collected simultaneously with the EMG signal and with the same sampling rate (2,000 Hz). The children were instructed to distribute their weight evenly between both limbs. The occurrence of the sudden drop event of one of the platform traps (extremity assessed) was communicated in advance, although the participants were unaware of the precise moment of the fall. To ensure isolation from environmental noise, the children wore headphones and eye masks. During the disturbance, the sEMG signals of the specified muscles were recorded. Three attempts were made and the average of them was used to obtain the muscle reaction time. The muscle reaction time was measured in milliseconds (ms) and determined when the sEMG activity exceeded a threshold of at least 3 standard deviations from the mean resting signal, which had a duration of 150 ms, and maintained this threshold for at least 25 ms. Data was analyzed using EMGworks Analysis 4.7.3 software (Delsys Inc., Boston, MA, USA).

Table 1 Sensor locations.

Muscles	Starting posture	Location	Orientation	
Rectus Femoris	Sitting on a table with the knees in slight flexion and the upper limbs slightly bends backward.	The electrodes need to be placed at 50% on the line from the anterior spina iliaca superior to the superior part of the patella.	In the direction of the line from the anterior spina iliaca superior to the superior part of the patella.	
Vastus Medialis	Sitting on a table with the knees slightly flexed and the upper limbs slightly bent backward.	Electrodes need to be placed at 80% on the line between the anterior spina iliaca superior and the joint space in front of the anterior border of the medial ligament.	Almost perpendicular to the line between the anterior spina iliaca superior and the joint space in front of the anterior border of the medial ligament.	
Vastus Lateralis	Sitting on a table with the knees in slight flexion and the upper limbs slightly bends backward.	Electrodes need to be placed at 2/3 on the line from the anterior spina iliaca superior to the lateral side of the patella.	In the direction of the muscle fibers.	

Statical analyses

Data were analyzed with Graph Pad Prism 9.0 statistical software (GraphPad Software, La Jolla, CA, USA). The data of the studied sample are presented as mean and standard deviation for continuous variables and as percentages for categorical variables. The Shapiro–Wilk test was performed to determine the distribution of the data. To compare muscle reaction times according to BMI and fat mass percentage, the t-student test was used for independent samples. A multiple linear regression model (95% confidence interval) was applied. The dependent variable was muscle reaction time, while the independent variables were BMI or fat mass percentage and sex (boys and girls). The regression models were made separately in two ways: (a) according to the classification by BMI (normal-weight and overweight/obese); (b) according to the classification by fat mass percentage (normal and overfat/obese). Two regression models were generated; model 1 included BMI or fat mass percentage and sex, while model 2 only considered BMI or fat mass percentage. The goodness of fit was assessed using the R2 coefficient. A collinearity diagnosis was conducted for each variable included in the regression models. Variables with tolerance values less than 0.10 and variance inflation factor (VIF) values exceeding 10.0 were eliminated. The level of significance for all statistical tests was set at <0.05.

Results

Of the forty-two children participated in the study (54.5% girls), 57.1% were classified as overweight/obese according to BMI, and 42.9% were classified as overfat/obese according to fat mass percentage.

Normal-weight children (age: 11.68 ± 0.33 years; body mass: 39.34 ± 3.49 kg; height: 1.48 ± 0.06 m) exhibited a muscle reaction time of 85.9 ± 18.5 ms for the rectus femoris, 79.1 ± 9.36 ms for the vastus medialis, and 81.4 ± 8.1 ms for the vastus lateralis. Conversely, overweight/obese children (age: 11.53 ± 0.45 years; body mass: 53.64 ± 10.17 kg; height: 1.49 ± 0.06 m) showed a longer muscle reaction time with values of 104.0 ± 34.1 ms for the rectus femoris, 84.5 ± 24.7 ms for the vastus medialis, and 95.9 ± 28.4 ms for the vastus lateralis. Upon classifying participants based on their percentage of body fat, children with normal fat levels (age: 11.70 ± 0.43 years; body mass: 40.26 ± 4.31 kg; height: 1.46 ± 0.06 m) exhibited the following muscle reaction times: 84.2 ± 16.3 ms for the rectus femoris, 79.8 ± 19.6 ms for the vastus medialis, and 89.7 ± 20.9 ms for the vastus lateralis. Contrariwise, children categorized as overfat/obese (age: 11.44 ± 0.37 years; body mass: 57.82 ± 9.00 kg; height: 1.51 ± 0.05 m) showed longer muscle reaction times, measuring 97.0 ± 28.7 ms for the rectus femoris, 85.2 ± 19.7 ms for the vastus medialis, and 104.9 ± 35.1 ms for the vastus lateralis. Figure 1 graphically shows the muscle reaction time results according to both classifications. When children were classified by BMI, significant differences were observed between normal-weight vs. overweight/obese for the reaction times of the rectus femoris (p = 0.048) and vastus lateralis (p = 0.042) muscles. Likewise, when children were compared according to percentage of fat, significant differences were observed between normal fat vs. overfat/obese for the reaction times of the rectus femoris muscles (p = 0.047) and vastus lateralis (p = 0.037).

Figure 1 Comparison of the reaction time of the rectus femoris (RF), vastus medialis (VM), and vastus lateralis (VL) muscles according to BMI (A) and fat mass (B).

When BMI and body fat classified the children, the muscle reaction time was higher in the overweight/obese and overfat/obese groups, respectively. *p < 0.05.

Table 2 shows the linear regression models obtained for muscle reaction times based on BMI. The models were significant for the rectus femoris (R2 = 0.21; p = 0.044) and vastus lateralis (R2 = 0.22; p = 0.042) muscles. In both analyses, it was observed that sex did not influence muscle reaction times. Therefore, model 2 was the one that best represented the influence of BMI on muscle reaction time, showing that overweight/obese children have a longer muscle reaction time for both the rectus femoris (β = 18.13; p = 0.048) and the vastus lateralis (β = 14.51; p = 0.042).

Table 2 Multiple linear regression models obtained for muscle reaction time according to BMI.

Muscles	R2	Coefficient β	P value	95% CI	
Rectus femoris						
Model 1	0.20		ns			
Intercept		83.93	ns	68.00	99.86	
Overweight/obese		18.25	ns	0.08	36.43	
Boys		4.47	ns	−13.71	22.64	
Model 2	0.21		0.044			
Intercept		85.91	0.000	72.32	99.50	
Overweight/obese		18.13	0.048	0.15	35.11	
Vastus medialis						
Model 1	0.02		ns			
Intercept		80.15	ns	69.14	91.16	
Overweight/obese		5.33	ns	−7.23	17.89	
Boys		−2.35	ns	−14.92	10.21	
Model 2	0.02		ns			
Intercept		79.11	ns	69.72	88.49	
Overweight/obese		5.39	ns	−7.02	17.80	
Vastus lateralis						
Model 1	0.21		ns			
Intercept		79.04	ns	66.69	91.40	
Overweight/obese		14.66	ns	0.57	28.75	
Boys		5.4	ns	−8.68	19.49	
Model 2	0.22		0.042			
Intercept		81.44	0.000	70.86	92.03	
Overweight/obese		14.51	0.042	0.50	28.51	
Notes.

95% CI 95% Confidence Interval

ns no significant

The reference for the overweight/obese category was the normal weight category. The reference for boys was girls.

Table 3 shows the linear regression models obtained for muscle reaction times based on fat mass percentage. Model 2 showed significant findings for the rectus femoris muscle (R2 = 0.20; p = 0.045), whereas both model 1 (R2 = 0.18; p = 0.019) and model 2 (R2 = 0.23; p = 0.018) yielded significant results for the vastus lateralis muscle. In both models, it was observed that sex did not influence muscle reaction times. Therefore, model 2 was the one that best represented the influence of fat mass percentage on muscle reaction time, showing that overfat/obese children have a longer muscle reaction time for both the rectus femoris (β = 18.24; p = 0.046) and the vastus lateralis (β = 16.57; p = 0.018).

Table 3 Multiple linear regression models obtained for muscle reaction time according to fat mass percentage.

Muscles	R2	Coefficient β	P value	95% CI	
Rectus femoris						
Model 1	0.10		ns			
Intercept		82.09	ns	66.08	98.10	
Overfat/obese		21.41	ns	2.72	40.10	
Boys		10.50	ns	−8.30	29.30	
Model 2	0.20		0.045			
Intercept		88.02	0.000	76.01	99.98	
Overfat/obese		18.24	0.046	0.38	36.09	
Vastus medialis						
Model 1	0.06		ns			
Intercept		77.51	ns	66.52	88.51	
Overfat/obese		9.84	ns	−2.98	22.68	
Boys		0.50	ns	−12.40	13.42	
Model 2	0.06		ns			
Intercept		77.80	ns	69.68	86.92	
Overfat/obese		9.69	ns	−2.37	21.76	
Vastus lateralis						
Model 1	0.18		0.019			
Intercept		75.98	0.000	63.90	88.06	
Overfat/obese		19.92	0.006	5.81	34.02	
Boys		11.08	ns	−3.11	25.26	
Model 2	0.23		0.018			
Intercept		82.24	0.000	73.04	91.44	
Overfat/obese		16.57	0.018	2.86	30.25	
Notes.

95% CI 95% Confidence Interval

ns no significant

The reference for the Overfat/obese category was the normal fat category. The reference for boys was girls.

Discussion

Our results indicate BMI and fat mass classification negatively alter the muscle reaction times in children. Overweight/obese or overfat/obese children showed longer reaction times in the rectus femoris and vastus lateralis muscles, irrespective of their sex, compared to children with normal weight. Previous studies have also reported altered neuromuscular gait patterns in obese children (Blakemore et al., 2013; Hills & Parker, 1993) and adults with excess weight (Amiri et al., 2015; Bollinger & Ransom, 2020). Both studies conducted in children revealed that body mass affects muscle activity patterns in children’s gait. Children with higher body mass showed greater amplitude in sEMG signals and a longer duration of muscle activation compared with children with normal body mass (Blakemore et al., 2013; Hills & Parker, 1993). However, our study is the first to observe differences in sEMG reaction times in overweight/obese children. Therefore, the hypotheses are partially confirmed.

The main finding of this study indicates that overweight/obese and overfat/obese children exhibit a delayed muscle reaction time when confronted with a sudden destabilization task. One hypothesis that could explain this finding is linked to the greater amount of adipose tissue reported in overweight/obese children. In obese people, there is a chronic accumulation of adipose tissue, which leads to increased levels of circulating pro-inflammatory cytokines, such as tumor necrosis factor α (TNFα) and certain interleukins (e.g., IL-1α and IL-6) (Straight, Toth & Miller, 2021; Tomlinson et al., 2016; Uranga & Keller, 2019). These cytokines play a crucial role in cell signaling in response to both acute and chronic systemic inflammation. They may have a detrimental effect on skeletal muscle by stimulating muscle protein breakdown, which can result in impaired muscle function and performance (Addison et al., 2014; Straight, Toth & Miller, 2021; Tomlinson et al., 2016; Uranga & Keller, 2019). The accumulation of adipose tissue could also be related to the slowing of motor nerve conduction velocity reported in obese people (Majumdar et al., 2017). Therefore, changes at the muscle level and in nerve conduction pathways could be factors that support understanding the delay in muscle reaction times observed in overweight/obese and overfat/obese children in our study. Finally, another factor that may contribute to the changes induced by excessive weight on muscle activation patterns is arthrogenic muscle inhibition (AMI) of the quadriceps (Hopkins & Ingersoll, 2022). AMI refers to the phenomenon in which the quadriceps muscle activation is inhibited due to inflammation and/or joint edema (Hopkins & Ingersoll, 2022; Konishi, Yoshii & Ingersoll, 2022). In persons with obesity, the additional weight can impose increased stress on the joints, resulting in chronic inflammation and joint dysfunction (Hopkins & Ingersoll, 2022; Konishi, Yoshii & Ingersoll, 2022). Consequently, AMI could be an additional factor contributing to impaired motor function in the obese children examined in this study.

The changes at the muscular level are not the only ones known in the sensorimotor system. The scientific evidence additionally reveals alterations in the other two groups of this system: sensory (Saleh & Abd El-Hakiem Abd El-Nabie, 2018) and cortical (Li et al., 2022; Pan et al., 2022), which would also contribute to understanding the lower neuromuscular capacity of overweight/obese children detected in this study and the alterations in motor skills and functional performance observed in previous studies (Barros et al., 2022; Guzmán-Muñoz et al., 2023; Thivel et al., 2016). At the sensory level, it has been seen, specifically, that proprioception is decreased in obese children, which is considered a determining factor in the poor motor control observed in these persons (Saleh & Abd El-Hakiem Abd El-Nabie, 2018). Likewise, at the cortical level, a decrease in the volume of gray matter has been reported in overweight/obese people, not only in areas related to reward but also in areas of sensorimotor integration (Li et al., 2022; Pan et al., 2022), which suggests that the deficits in motor skills and functional performance that obese children present could also be associated with adverse changes at this level. Therefore, the negative effects of childhood obesity on physical function should be comprehensively addressed based on the sensorimotor system.

The delayed reaction times of the quadriceps muscles, which are associated with obesity, could pose a significant limitation in daily living, and potentially become a risk factor for musculoskeletal injuries. Obesity can make it excessively challenging for obese children to perform daily activities that require quadriceps contractions with a wide range of motion, such as kneeling and squatting (Tallis, James & Seebacher, 2018). As a result, their physical functioning becomes restricted. In our study, we specifically observed a significant delay in reaction time in the rectus femoris and vastus lateralis muscles of obese children. However, in the vastus medialis muscle, although there was a tendency towards a longer reaction time in overweight/obese children, the differences were not statistically significant compared to their normal-weight peers. This difference in reaction times could potentially be attributed to variations in the distribution of muscle fiber types observed in the muscles under examination. Both the rectus femoris and vastus lateralis muscles have been reported to possess a higher proportion of fast twitch (type II) muscle fibers than the vastus medialis (Johnson et al., 1973). In persons with obesity, there is often a decrease in the proportion of type II muscle fibers and an increase in the proportion of slow twitch fibers (type I) (Tanner et al., 2002). This shift in fiber composition may explain why the rectus femoris and vastus lateralis muscles exhibit more pronounced alterations in this sample.

This study’s limitations include the non-probabilistic participant selection and the small sample size. The latter hindered the classification of children into more BMI categories, such as normal weight, overweight, or obese, and instead required a binary categorization. Another limitation of the study was the lack of assessment of subcutaneous fat, which can on impact the acquisition of sEMG signals by acting as a low-pass filter. We recommend incorporating adjustment variables like physical activity level and muscle mass percentage to enhance future studies. Additionally, assessing specific intramuscular fat mass percentages could be achieved using techniques such as ultrasonography.

Conclusions

The BMI and fat mass classification negatively alter to quadriceps muscle reaction times in children. Irrespective of gender, children categorized as overweight/obese or overfat/obese demonstrated prolonged reaction times in the rectus femoris and vastus lateralis muscles in comparison to their peers with normal weight. Based on these findings, it is suggested that in overweight and obese children, efforts not only focus on reducing body weight, but that be complemented with training and/or rehabilitation programs that aim to modify body composition (decrease the percentage of body fat) and that promote motor stimulation to preserve the normal physiological function of the musculoskeletal system.

Supplemental Information

Supplemental Information 1 Muscle reaction times of children classified according to BMI

Supplemental Information 2 Muscle reaction times of children classified according to body fat percentage

Supplemental Information 3 STROBE Checklist

Additional Information and Declarations

Competing Interests

Author Contributions

Human Ethics

Data Availability

Guillermo Mendez-Rebolledo is an Academic Editor for PeerJ.

Eduardo Guzmán-Muñoz conceived and designed the experiments, performed the experiments, analyzed the data, prepared figures and/or tables, authored or reviewed drafts of the article, and approved the final draft.

Guillermo Mendez-Rebolledo conceived and designed the experiments, prepared figures and/or tables, authored or reviewed drafts of the article, and approved the final draft.

Sergio Sazo-Rodriguez performed the experiments, authored or reviewed drafts of the article, and approved the final draft.

Joaquín Salazar-Méndez performed the experiments, authored or reviewed drafts of the article, and approved the final draft.

Pablo Valdes-Badilla analyzed the data, authored or reviewed drafts of the article, and approved the final draft.

Cristian Nuñez-Espinosa performed the experiments, authored or reviewed drafts of the article, and approved the final draft.

Tomas Herrera-Valenzuela analyzed the data, authored or reviewed drafts of the article, and approved the final draft.

The following information was supplied relating to ethical approvals (i.e., approving body and any reference numbers):

The study received ethical approval from the local Ethics Committee (Universidad Santo Tomás, Chile), identified by registration number 13320.

The following information was supplied regarding data availability:

The raw measurements are available in the Supplementary Files.

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
