# Peer review of "Quadriceps muscle reaction time in obese children"

_PeerJ, doi:10.7717/peerj.17050_

## Round 0.1 · original submission · Major Revisions

Three reviewers have provided constructive and informative comments. The manuscript should be markedly improved if edited following the suggestions.

Reviewer 1 ·

Basic reporting

The authors have presented research examining the neuromuscular response of the quadriceps muscles to a perturbation between children classified as “normal” and/or “overweight/obese”/”overfat/obese”. There is merit is evaluating the response of the muscles to a perturbation and motor control is important for improved development as children age. However, there a numerous issues that the authors must address to better communicate their ideas. The following comments are provided to assist the authors in further appraisal/reflection of their ideas.
The increased reaction time of children who are classified as overweight/obese could be due to: 1) enhanced filtering of the tissues due to the increased adipose tissues, and 2) the greater body mass could increase moment of inertia requiring a latency response due to the longer time to perturb the system (i.e., more force is required to stimulate the system to respond). Further, the use of BMI to classify a group needs to be approached cautiously. The authors mentioned two groups: obese/overweight and overfat/obese. The authors do not make a distinction between these groups as there were different measures used to classify the participants.
The term “nutritional status” is misleading. The authors used BMI and bioelectrical impedance to identify children as “normal” or overweight/obese or overfat/obese – and these could be classified further, however these additional groups would reduce the statistical power for analysis.
The Methods section requires additional details (see specific comments below).
In the Results, were comparisons between Boys and Girls performed? It is not clear what benefit the two models presented for the overweight/obese and overfat/obese classifications provide. Based upon the data, there are no differences between boys and girls that were identified.
There are a number of limitations to this study, as mentioned by the authors in the Discussion. The activity level of the participants was not addressed and can be a determining factor in neuromuscular response.

Experimental design

Methods:
p.3 lines 118-121: what were the demographic data (height, mass, age) of each classification of children (normal, overweight/obese, and overfat/obese)?
p.4 lines 138-142: are the standard deviations based upon a mean of BMI for children at that age level? This needs further clarification
p.4 lines 143-147: the use of this instrument to measure fat percentage has been unclear. Generally, this instrument has overestimated the percentage of body fat in males and females (Pribyl et al. 2011, Int J Exer Sci 4(1); 93-101) – however, most studies of BMI have used adults and not children for validation (Vasold et al. 2019, Int J Sports Nutr Exer Met, 29, 406-410)
P4. Lines 158-160: “standard mode rejection ration” should be common mode rejection ratio (CMRR), and to the authors mean 0.75 microVolts RMS?
p.5 lines 165-169: was only the one platform tilted compared to the other? This is not clear. It is also not clear what the threshold was of the accelerometer signal used to measure the perturbation. Can it be assumed that the EMG and accelerometer data are synchronized? The authors indicate the EMG are collected using Delsys software, but do not reference the same software for the accelerometer. It is also not clear what the collection rate was for the accelerometer. If the rates of data collection are different between the EMG and accelerometer, were the data down sampled or up sampled?
p.5 lines 172-174: how was the “trap” controlled for the experiment? What was the displacement of the trap and what was the rate at which it fell? How did the author ensure that the participants maintained equal weight distributions of the platforms when they knew in advance one of the platforms would fall? It is possible that the participants anticipated the drop and adjusted their weight distribution accordingly.
p.5 lines 176-177: three trials of attempts were performed but only one trial was used for the evaluation. This seems to negate any with-subject variability that may be present. Was an algorithm used to determine the EMG threshold, or was this performed using visual inspection?
p.5 lines 189-190: the authors indicated that overweight/obesity levels would be the primary independent variables, but did not indicate that sex was also considered.
Table 1. for the vastus medialis electrode placement, it is unclear what “80%” refers to for the location of the sensor, likewise, what is 2/3 in reference to the location of the sensor placed over the vastus lateralis?

Validity of the findings

p.6 lines 201-203: using BMI, 24 of the 42 children were classified as overweight/obese, while using bioelectrical impedance 18 children were classified as overfat/obese. This indicates that 14% of the children are classified into one category and not the other. This calls into question the validity of the classification system used by the authors.

Annotated reviews are not available for download in order to protect the identity of reviewers who chose to remain anonymous.
Cite this review as

·

Basic reporting

Firstly, I would like to apologize for the delay in submitting this review. As I assume you will understand, often in our work, the initial plan gets derailed due to sudden emergencies that need to be addressed promptly. With that said, I will now proceed with my overall assessment of the manuscript.

The article titled "Quadriceps muscle reaction time in obese children" is easy to read and understand. In my opinion, although I am not qualified to evaluate it precisely, the English is sufficiently good. The references are also appropriate and up-to-date, although I suggest conducting a more thorough review of the references used, especially in the second paragraph of the introduction (to highlight the issues that obesity can cause in terms of motor skills, particularly balance and gait). Likewise, in line 62, there is a reference in Vancouver format when APA format has been used throughout the rest of the document.

The structure of the work is suitable, as are the figures and tables. Regarding the data provided by the authors (raw data of the manuscript), the gender of the participants is missing. Additionally, I suggest including the variables Fat % and BMI with their original values, not just the categories. This request is related to the need to modify some statistical analyses (see in the following sections).

Experimental design

Overall, the article sufficiently describes the problem to be investigated, as well as the objectives and hypotheses. It might be a good idea in the last paragraph of the introduction to justify why it is possible that children may exhibit different effects than adults in terms of the influence of obesity on reaction time. On the other hand, there are some aspects related to the description of the methodology and statistical analyses that could be improved. Please consider the following aspects:
• The authors group BMI and Fat% as indicators of nutritional status. However, nutritional status can encompass many other factors that have not been assessed. I would be more specific and refer to weight status and body composition.
• In calculating the sample size, provide Cohen's d instead of the percentage change used in the analyses. Furthermore, the citation of De Carvalho seems relevant for the introduction, and it did not appear to me that it was included.
• Include variables of interest that could act as confounding factors. That is, when describing the sample, create a table for the groups being compared and analyze whether there are differences between the groups in variables such as gender, age, physical activity... Any information the authors can provide will be welcomed.
• Were the ACC and EMG variables synchronized to calculate reaction time? How was sincronization done?
• Did the authors somehow verify that the weight distribution was the same between the two platforms? If participants realize that only one platform is rotating, they may redistribute their weight so that a larger proportion is on the rotating platform. I understand that participants were instructed to distribute weight evenly, but was this checked in any way?
• Why was one of the three attempts chosen randomly? This seems a bit unusual since the most common practice is to use the average of the three attempts.
• What was the tilt speed of the platform?
• Has the possibility been considered that body fat acting as a filter for electromyographic activity may hinder the detection of the onset of muscle action and contribute to an increase in reaction times for the overweight/obese group?
• It appears that the authors have used linear regression models with categorical predictors. I strongly recommend using the original variables on a quantitative scale.
• In the last sentence of the results, the beta coefficients are the same as in line 224. Is it possible that this is an error?

Validity of the findings

Overall, the article is methodologically robust, which helps us consider its validity to be adequate. However, in the previous section, I have raised some issues that need to be addressed to ensure the validity and impact of the results.

Cite this review as

Reviewer 3 ·

Basic reporting

Overall it is easy to read. There are only a few grammatical errors.
Line 62, has a citation formatting error.
My biggest issue is with the term "nutritional status" as there was no measurement of nutrition. I believe a more correct term would be "weight status" since they are looking at BMI. Please fix throughout the manuscript.

Experimental design

I am not sure why they classify as 'normal', 'overweight' and 'obese', but then group into normal and overweight/obese? It is not clear why the groupings were that way. Also, did they not have any who were underweight?
Also, has the research compared surface EMG to wire EMG to see if the data is reliable in obese/overweight people? I feel like this should be addressed.

Validity of the findings

Also, has the research compared surface EMG to wire EMG to see if the data is reliable in obese/overweight people? I feel like this should be addressed to help determine the validity of the findings.
I think looking at the relationships among normal, overweight, and obese may give a more valid picture, but maybe this was not possible with the statistics.

Additional comments

Line 55: what type of efficiency?
Line 60, grammar issue in the sentence "In this context,...
Line 68: are muscle RT and absolute latency the same thing? unclear as written
Line 73-74: abrupt transition
Line 90: no citations for "Few studies.."
Line 111: grammar issues. are they only obese when walking?
Line 134: what was considered appropriate dress?
Line 142, 151: were any underweight or underfat?
Line 176: why was 1 random trial used and not the average or median?
Line 191: why did you group into 2 categories when you had 3? normal, overweight, and obese?
Line 201: should read as "Of the forty-two children who participated...
Line 235: how were they related? correlated, inversely correlated? be more specific
Line 238: what type of altered patterns were reported?
Discussion: Was there a decrease in RT of the muscle firing, or a delayed signal due to the increased fat tissue? Please address.
Have any studies compared the results of surface EMG with wired?

Table 1: Starting posture: what is "slight flexion" - add degrees, where are the feet positioned? can you add a small drawing for each? Location: distal or proximal placement, be specific

Cite this review as

---

## Round 0.2 · accepted · Accept

The reviewers and I thank you for a successful revision.

Reviewer 1 ·

Basic reporting

Introduction - this information provides good support in the literature for the purpose and hypothesis of the authors
Methods - The methodology is presented clearly- were both limbs monitored or only one? This would provide further information regarding anticipation of the participants to the perturbation
Results - These are presented well
Discussion - this section incorporates the results and ideas of current literature, while also noting the limitations of the equipment

Experimental design

What other controls were initiated to reduce the probability of the participants anticipating the drop?

Validity of the findings

The finding seem to be valid, based upon the information provided

Cite this review as

Reviewer 3 ·

Basic reporting

no comment

Experimental design

no comment

Validity of the findings

no comment

Additional comments

Thank you for addressing my comments.

Cite this review as